# Effect of Different ET-Based Irrigation Scheduling on Grain Yield and Water Use Efficiency of Drip Irrigated Maize

**Dejan Simić [1], Borivoj Pejić [1], Goran Bekavac [2], Ksenija Mačkić [1,*], Bojan Vojnov [1], Ivana Bajić [2] and Vladimir Sikora [2]**

1   Faculty of Agriculture, University of Novi Sad, 21000 Novi Sad, Serbia; dejansymic@yahoo.com (D.S.); borivoj.pejic@polj.uns.ac.rs (B.P.); bojan.vojnov@polj.uns.ac.rs (B.V.)
2   Institute of Field and Vegetable Crops Novi Sad, 21000 Novi Sad, Serbia; goran.bekavac@nsseme.com (G.B.); ivana.bajic@ifvcns.ns.ac.rs (I.B.); vladimir.sikora@ifvcns.ns.ac.rs (V.S.)
*   Correspondence: ksenija.mackic@polj.uns.ac.rs

**Abstract:** The development of irrigation schedules based on water balance implies a study of daily plant water requirements. A properly selected irrigation method is also of most importance. The objective of this study was to find out how surface drip irrigation (SDI) and shallow subsurface drip irrigation (SSDI), as well as different ET-based irrigation scheduling for maize (reference evapotranspiration ($ET_o$), pan evaporation ($E_o$), and local climatic coefficients ($l_c$)), affect grain yield, water use efficiency (WUE), and yield response factor ($K_y$) of maize. The field experiments were conducted in Vojvodina, a northern part of the Republic of Serbia, on the calcareous gleyic chernozem soil, using a complete block design in three replicates in 2019–2021. The water balance method was used for irrigation scheduling. The nonirrigated treatment was used as a control. The yield in irrigation conditions was statistically higher as compared with the nonirrigated control variant. Concerning the tested parameters, especially the maize yield, reference evapotranspiration ($ET_o$) should be recommended as the most acceptable method for assessing maize evapotranspiration. Preference should be given to SSDI compared to SDI because the installation of laterals can be performed together with the sowing, which can ensure the uniform and timely emergence of plants. Based on the $K_y$ coefficient of 0.71, it can be concluded that maize is moderately tolerant to water stress in Vojvodina's temperate climate. The results can contribute to precise planning and efficient irrigation of maize in the region, implying high and stable yields.

**Keywords:** maize; evapotranspiration; drip irrigation; yield; water use efficiency

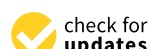



## 1. Introduction

The production of maize (*Zea mays* L.) has a significant place in world agriculture, with a production potential of approximately 1162 M t harvested and 197 M ha planted area with an average yield of 5.8 t ha$^{-1}$ [1], making it the second most widely grown crop in the world after wheat. Maize is the most important crop in Serbia, providing the highest economic revenue. Over the last three years (2019–2021), in Serbia, maize was grown on approximately 1 M ha with a total grain production of 7.1 M t and an average yield of 7.1 t ha$^{-1}$. Approximately 560,000 ha were devoted to maize in the northern Serbian province of Vojvodina, with an annual production of 4.4 M t and an average yield of 7.9 t ha$^{-1}$ [2]. In Vojvodina, maize is mainly produced under rainfed conditions. However, high and stable maize yields can only be achieved by supplementing the crop's water needs through irrigation in the variable climatic conditions of Vojvodina, in which summers are semi-arid to arid [3].

A reliable estimation of plant water requirements is essential for agricultural planning and efficient management of irrigation systems. The water requirements of maize in the Vojvodina region vary from 460 to 540 mm depending on the length of vegetation of the

hybrid grown [4]. Due to unplanned rainfall and its distribution during the growing season, irrigation in Vojvodina can be considered a rainfall supplement for successful agricultural production [5]. Various studies conducted in different climatic and soil conditions indicate that irrigation can have a significant impact on maize productivity [6,7] and that for high yields of maize, an adequate water supply is required. Thus, irrigation scheduling is an essential aspect of irrigation water management. Several methods are used to determine the irrigation time, the most common being the water balance method, because of its simplicity and reliability. Precise calculation of daily crop evapotranspiration ($ET_d$) is required to schedule irrigation using the water balance method, [8]. Doorenbos and Pruitt [9] suggested the determination of $ET_d$ through reference evapotranspiration ($ET_o$) and crop coefficients ($k_c$). The influence of climatic factors on $k_c$ is limited, allowing the acceptability of this approach for different locations and climatic conditions. Allen et al. [10] recommended the Penman–Monteith method (FAO 56 PM) as a global standard for $ET_o$ calculation. Due to its simplicity and similarity with the obtained data using the Penman–Monteith method, the Serbian Hydrometeorological Institute presents daily $ET_o$ values calculated using the Hargreaves method [11]. $ET_o$ can be calculated from pan evaporation ($E_o$) and a pan coefficient ($K_p$) [12,13]. To calculate $ET_d$ from $E_o$, the plant coefficients (k) should be determined [14,15]. For local climatic and soil conditions, crop water requirement was determined using the hydrophytothermal indexes [14,16,17]. They show how many millimeters of water plants spend on evapotranspiration for each degree of mean daily air temperature. This method of calculating plant evapotranspiration is successfully applied in irrigation scheduling in Vojvodina.

Irrigation scheduling involves choosing the correct irrigation method. Recently, in the region, surface (SDI) and subsurface (SSDI) drip irrigation have been gaining more importance due to their numerous advantages. Drip irrigation is considered the most efficient form of irrigation compared with other irrigation methods. This is due to the delivery of water directly to the plant root zone [18], minimizing evaporation [19], and deep percolation beneath the plant root zone [20]. SSDI lessens wind drift and overspray and prevents crust formation, hindering soil aeration and rainwater infiltration [21], vandalism, and animal damage. SSDI is a relatively new technological innovation. Wang et al. [22] defined SSDI as when the drip lateral depth is less than 10 cm below the soil surface. There are very few results in the literature regarding subsurface irrigation with laterals placed shallow in the soil and removed from the plot before harvest and used in subsequent years. Samardžić et al. [23] pointed out that preference should be given to SSDI irrigation with regard to SDI, as placing laterals can be performed together with the sowing or planting of plants, which can affect the uniform and timely emergence of plants. Wang et al. [22] reported some other advantages of SSDI such as low installation cost, convenient mainte- nance, and renewal.

The crop response factor ($K_y$) quantifies the crop's sensitivity to water stress as the amount of yield (Y) lost per unit of evapotranspiration (ET) loss. A higher $K_y$ value implies higher yield losses due to water deficit [9]. Precise determination implies a suf- ficient range and number of data for Y and ET and their linear relationship [5,24,25]. Doorenbos and Kassam [9] reported a $K_y$ value for maize of 1.25 for the whole growing period. Payero et al. [26] estimated $K_y$ values of 0.4, 1.5, 0.5, and 0.2 for vegetative, flowering, yield formation, and ripening stages, respectively, indicating the highest sen- sitivity to water stress during the flowering stage. Pejić et al. [5] concluded that maize is moderately sensitive to soil water stress ($K_y$ = 0.54) in the temperate climatic condi- tions of the Vojvodina region. The least susceptible stage is vegetative ($K_y$ = 0.37) then grain filling and maturity ($K_y$ = 0.41), and the most sensitive is flowering and pollination ($K_y$ = 0.52). If water is limited, farmers should always irrigate plants at the most sensitive stage. Kobossi and Kaveh [27] suggested $K_y$ values for the entire growing season rather than for individual growing stages as the decrease in yield due to water stress during specific periods, such as vegetative and ripening periods, is relatively small compared to the large decrease in the yield formation period. Water scarcity has become a major

concern affecting sustainable agricultural production. The competition for water resources is increasing to a large extent as irrigated activities account for almost 70% of global water withdrawals, reaching as much as 95% in some developing countries [28], and because of that, irrigation should be used as efficiently as possible. Irrigation water use efficiency (IWUE) is defined as the amount of yield produced per unit of irrigation water applied ($kg\,m^{-3}$). It is a valuable indicator for quantifying the impact of irrigation scheduling decisions and irrigation water applied on crop yield. IWUE values are not optimal if the irrigation schedule is not synchronized with crops' water needs, soil properties, and weather conditions [29]. Howell [30] pointed out that IWUE tends to increase with a decline in irrigation but without a deficit of water in any single growth period of plants. Evapotranspiration water use efficiency (ETWUE) determines whether the growing season is suitable for crop production. ETWUE primarily depends on the amount and distribution of rainfall. All factors that increase yield and reduce water use from ET are beneficial for ETWUE [31]. Pejić et al. [32] emphasized that special attention should be paid when comparing WUE's results. In climatic conditions with supplemental irrigation, WUE calculations differ from those in arid regions where crop production cannot be accomplished under conditions of the plant's natural water supply. They also indicated that it is important to know in which units the results are expressed ($kg\,m^{-3}$, $t\,ha^{-1}\,mm$, or $g\,L^{-1}$).

This study aimed to determine the effects of surface and shallow subsurface drip irrigation and different ET-based irrigation scheduling programs on grain yield, water use efficiency, and yield response factor of maize. The results obtained will provide experts with useful information on the practical possibilities of drip irrigation both on the surface and subsurface and make recommendations on how to properly water maize, bringing high and stable yields. Furthermore, the most acceptable method for assessing the daily evapotranspiration of maize will be proposed during the planning, design, and operation of irrigation systems.

## 2. Materials and Methods

### 2.1. Site Description

The field experiments were conducted in the experimental field of the Institute of Field and Vegetable Crops Novi Sad at the Department of Alternative Plant Species in Bački Petrovac (45°19′ N latitude, 19°50′ E longitude, and 84 m above sea level) in 2019 and were repeated in 2020 and 2021. The climate is moderately continental, with four marked seasons [33]. Over the 1984–2018 period, the annual mean values of the air temperature, precipitation, and relative humidity were 12.1 °C (19.3 °C in the growing season from April to September), 626 mm (328 mm or approximately 50% in the growing season), and 77% (72.7% in the growing season). According to the Hargreaves climate classification system, the study area is classified as semi-arid to arid during the summer period, from June to August [3]. Plants, therefore, required irrigation during the summer season to avoid drought stress. The source of irrigation water was a deep well and the water quality was classified as $C_2S_1$, with a pH value of 8.03 and an electrical conductivity of 0.8 dS m$^{-1}$.

### 2.2. Soil Properties

The soil of the experimental field belongs to the clayey texture type according to the Tommerup classification [34] and calcareous, gleyic chernozem according to the IUSS WRB working group. The physical and chemical properties of the soil and soil moisture characteristics are presented in Table 1. In terms of the above parameters, this soil is suitable for all types of crops and irrigation systems.

**Table 1.** Some physical and chemical properties of the soil and soil moisture characteristics in the experimental site.

| | |
|---|---|
| Soil depth (m) | 0.4 |
| Textural Status, (Sand/Clay/Silt) (%) | 41/34/25 |
| Soil Water Capacity (33 kPa) (mas. %) | 26.93 |
| Lento Capillary Moisture (625 kPa) (mas. %) | 16.61 |
| Wilting Point (1500 kPa) (mas. %) | 12.65 |
| Soil Bulk Density (g cm$^{-3}$) | 1.30 |
| Specific Gravity (g cm$^{-3}$) | 2.66 |
| Total Porosity (vol. %) | 49.13 |
| Readily Available Water (mm) | 54.5 |
| pH (KCl/H$_2$O) | 7.28/8.17 |
| Carbonates, CaCO$_3$ (%) | 6.01 |
| Organic matter, Humus (%) | 2.9 |
| N (%) | 0.19 |
| P$_2$O$_5$ (mg 100 g$^{-1}$) | 29.77 |
| K$_2$O (mg 100 g$^{-1}$) | 30.43 |

### 2.3. Crop Management, Experimental Design, and Irrigation Treatments

All recommended agronomic practices were applied for maize cultivation at the experimental site. The following field operations were conducted: Plowing at 0.3 m depth, seedbed preparation with a seedbed cultivator, and sowing with a pneumatic drill. In all three years of the experiment, according to recommendations based on the results of soil analysis, 450 kg ha$^{-1}$ of 15:15:15 NPK fertilizer (67.5 kg ha$^{-1}$ of N, K$_2$O, and P$_2$O$_5$) was applied to the experimental plots before plowing in the autumn, while 50 kg ha$^{-1}$ of ammonium sulfate—(NH$_4$)$_2$SO$_4$, which contains 21% nitrogen and 24% sulfur was added in the spring, in the 7–8 leaf stage, by top dressing. The preceding crop was fiber hemp, winter wheat, and grain sorghum in the first, second, and third years, respectively. Maize was sown on the 24, 23 April, and 4 May (seeds were sown 0.04–0.05 m deep, the interrow spacing was 0.70 m and the spacing between plants in a row was 0.19 m, giving 75,000 plants per hectare) using a 4-row pneumatic drill Majevica 4RK (Majevica, Bačka Palanka, Serbia), and manually harvested on 27 August, 10 September, and 5 September (at the stage of physiological maturity) in 2019, 2020, and 2021, respectively. Weed control was performed by inter-row cultivation and manual hoeing. Maize hybrid NS 3023, FAO 390, created at the Institute of Field and Vegetable Crops from Novi Sad, was used for the trials. The experiment was set up as a complete block design and repeated three times. The size of the experiment unit was 10.0 m$^2$ (1.4 m × 7.15 m).

The first factor was the irrigation method: Surface drip irrigation (SDI, S$_1$) and shallow subsurface drip irrigation (SSDI, S$_2$) (Figure 1). The second factor was different ET-based irrigation scheduling: Reference evapotranspiration (I$_1$), pan evaporation (I$_2$), and local bioclimatic method (l$_3$). I$_o$ was the control nonirrigated treatment.

Daily water use on evapotranspiration (ET$_d$) was computed by Equations (1)–(3).

$$ET_d = ET_o \times k_c \tag{1}$$

$$ET_d = E_o \times K_p \times k \tag{2}$$

$$ET_d = l_c \times t \tag{3}$$

where ET$_o$ is the reference evapotranspiration (mm), k$_c$ is the crop coefficient, E$_o$ is the pan evaporation (mm), K$_p$ is the pan coefficient (equal to 0.80 in the semiarid environment with an average air humidity of 40–70%, low wind speed, fetch 1 m), k is the plant coefficient, l$_c$ is hydrophytothermal indexes, and t is the mean daily air temperature (°C).

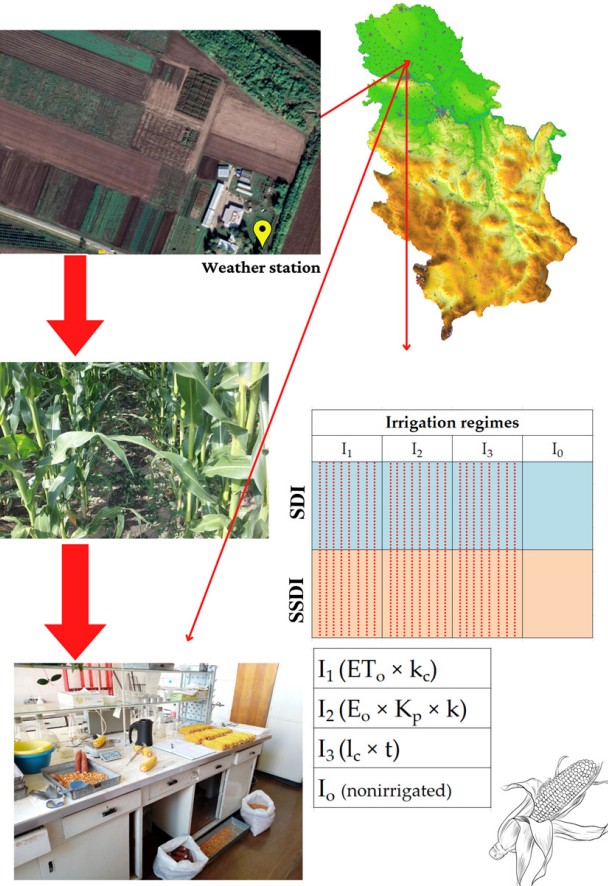

**Figure 1.** Location of the trial and experimental design (complete block) showing the main analyzed treatments: Surface drip irrigation (SDI) and shallow subsurface drip irrigation (SSDI), ET-based irrigation scheduling ($I_1$, $I_2$, $I_3$), and control, nonirrigated treatment ($I_o$).

Values of coefficients used for calculating maize evapotranspiration are presented in Table 2.

**Table 2.** Values of coefficients for calculating maize evapotranspiration.

| Months | $k_c$ | k | $l_c$ |
|---|---|---|---|
| May | <15.1 °C—0.3<br>15.1–18.3 °C—0.4<br>>18.3 °C–0.5 | 0.42 | <15.1 °C—0.12<br>15.1–18.3 °C—0.14<br>>18.3 °C—0.17 |
| Jun | <18.4 °C—0.7<br>18.4–21 °C—0.8<br>>21 °C—0.85 | 0.75 | <18.4 °C—0.14<br>18.4–21 °C—0.16<br>>21 °C—0.18 |
| July | <20.1 °C—1.05<br>20.1–22.7 °C—1.1<br>>22.7 °C—1.2 | 0.70 | <20.1 °C—0.16<br>20.1–22.7 °C—0.18<br>>22.7 °C—0.20 |
| August | <19.2 °C—0.8<br>19.2–22.4 °C—0.85<br>>22.4 °C—0.9 | 0.67 | <19.2 °C—0.15<br>19.2–22.4 °C—0.18<br>>22.4 °C—0.21 |
| September | <15.4 °C—0.5<br>15.4–18.3 °C—0.55<br>>18.3 °C—0.6 | 0.63 | <15.4 °C—0.10<br>15.4–18.3 °C—0.12<br>>18.3 °C—0.14 |

Daily values of $ET_o$ calculated by the Hargreaves equation [11] were taken from the Weather Service of the Republic of Serbia website. $E_o$ was measured daily using a Class A pan located at the weather station near the experimental plot, and mean daily air

temperature and rainfall. For local climatic and soil conditions, $ET_d$ was calculated using the hydrophytothermal indexes ($l_c$).

Irrigation scheduling was determined using the soil water balance that includes meteorological, soil, and crop data for a daily estimation of readily available water (RAW) in the effective root zone. The balance method estimates water depletion from the crop root zone due to the plant's evapotranspiration, with irrigation and effective rainfall as inputs.

Maize was irrigated by a drip irrigation system (PoliDrip Light PC 2.0 L h$^{-1}$ D16 33 cm, Poliext, Hungary). Laterals were placed in the middle of each row (0.7 m), on the surface of the soil (SDI), and buried under the surface at a depth of 0.05–0.06 m (SSDI). Drippers were spaced every 0.33 m with an average flow of 2.0 L h$^{-1}$ under an operating pressure of 0.1 MPa. Each plot had a valve to control irrigation. Irrigation started when RAW in the soil layer of 0.4 m was completely absorbed by plants. The irrigation depth was restricted to the soil depth of 0.4 m, where most maize roots are expected to grow (the effective crop root zone) [35]. The irrigation rate was 30 and 40 mm at the beginning and in the middle of the season, respectively. Runoff and capillary rise were assumed to be negligible, but in the case of heavy rain, greater than the capacity of the soil for RAW in a layer of 0.4 m, percolated water below the active root zone of maize was calculated. Flow and pressure gauges in the irrigation hose nozzles controlled the amount of irrigation water and the pressure in the system. The initial content of RAW in the soil layer of 0.4 m at the beginning of the growing season was detected by the gravimetric method. The soil was sampled by the destructive sampling method using a soil auger. Results were observed identically for all irrigation treatments, indicating uniform soil water distribution in the field from winter and spring precipitation.

$$RAW = 100 \times h \times a \text{ (Soil Water Capacity, 33 kPa} - \text{Lento Capillary Moisture, 625 kPa)} \quad (4)$$

where RAW is the readily available water (m$^{-3}$ ha$^{-1}$, mm), h is the depth of the soil (m), and a is the soil bulk density (g cm$^{-3}$).

The drip irrigation system's running time (RT) was computed based on Equation (5).

$$RT = \frac{\text{the volume of water applied}}{\text{number of drippers} \times \text{dripper discharge rate}} \quad (5)$$

where the volume of water applied is in l m$^{-2}$ and the dripper discharge rate is in L h$^{-1}$.

### 2.4. Sampling and Laboratory Analyses

Eight rows of maize were sown on all irrigation treatments and the control, nonirrigated treatment. The six middle rows were used for yield determination at harvest, while the first and last rows served as border lines. The yield (t ha$^{-1}$), adjusted to 14% moisture content, was computed based on the yield measured at the experimental unit. The moisture content of the grains was measured by a Digital grain moisture analyzer (mini GAC®plus, DICKEY-john®, Auburn, IL, USA). Analysis of yield components (weight of ear (g), number of grains per ear, weight of grains per ear (g), and weight of 1000 grains (g)) was performed on ten ears in three repetitions. The weight of 1000 grains was measured for each treatment and adjusted to 14% moisture content. Maize phenology was visually observed throughout the growing season.

### 2.5. Data Analyses

To evaluate the effects of surface and shallow subsurface drip irrigation and different ET-based irrigation scheduling on maize yield, irrigation water use efficiency (IWUE) and evapotranspiration water use efficiency (ETWUE) were calculated. IWUE and ETWUE were estimated according to Irmak et al. [20] and Bos [36], Equations (6) and (7).

$$IWUE = (Y_m - Y_a)/I \quad (6)$$

$$ETWUE = (Y_m - Y_a)/(ET_m - ET_a) \tag{7}$$

where $Y_m$ is the yield under irrigation treatment (kg ha$^{-1}$), $ET_m$ is the evapotranspiration (mm) corresponding to $Y_m$, $Y_a$ is the yield in nonirrigated treatment (kg ha$^{-1}$), $ET_a$ is the actual evapotranspiration (mm) corresponding to $Y_a$, and I is the seasonal irrigation water applied (m$^{-3}$ ha).

To express the yield lost per unit of evapotranspiration loss, the yield response factor ($K_y$) was computed according to Doorenbos and Kassam [37], Equation (8).

$$K_y = [1 - \frac{Y_a}{Y_m}]/[1 - \frac{ET_a}{ET_m}] \tag{8}$$

### 2.6. Statistical Analysis

The analysis of variance (ANOVA) was used to statistically analyze the data using the IBM SPSS Statistics software (version 26.0, modified 2021). Means were compared using Fisher's protected least significant difference (LSD) test for a 95% level of probability to identify significant changes between the treatments for yield and yield components of maize.

## 3. Results

### 3.1. Weather Conditions and Applied Irrigation Water Amount

Weather data were obtained from an on-site weather station located near the experimental plot (Figure 2). Whether the growing season is favorable for maize production in the region, the only authoritative is the comparison of rainfall and air temperature with the long-term average (LTA) for the period May-August, bearing in mind that maize was sown by mid of April, while physiological maturity was in the first half of September. The monthly values of the mentioned parameters are particularly important, as they are the daily extremes. In 2019, the growing period for maize lasted 126 days (Table 4). In that period, there was 313 mm of rainfall, 51 mm more than the LTA (262 mm) (Table 4, Figures 2 and 3). In the period from planting to the 4 July, 226.5 mm of rainfall fell, and hence, there was no need for irrigation in that part of the growing season. All irrigation events were performed from 5 July to 16 August (Figure 3). The amount of water added by irrigation was 150 mm, 110 mm, and 110 mm on the $I_1$, $I_2$, and $I_3$ variants, respectively (Figure 3, Table 4). In the growing season in 2019, the mean air temperature was 20.1 °C, which is 0.9 °C higher than the LTA (20.5 °C). Rainfall of 137.7 mm and 34.5 mm in May and July and temperature of 24.4 °C in August are particularly noteworthy (Figure 2). In 2020, the maize growing season lasted 141 days (Table 5); there was 370 mm of rainfall (Table 5, Figures 2 and 4). All irrigation was carried out from 9 to 31 July, with irrigation rates of 80 mm, 40 mm, and 80 mm on the $I_1$, $I_2$, and $I_3$ variants, respectively (Figure 4). In the period of May–August, there was 356.6 mm of rain, which was 94.6 mm more than the LTA. The mean air temperature was 21 °C or 0.5 °C higher than the LTA (Figure 2). Rainfall of 122.2 mm, 126.8 mm, and 42.9 mm in June, August, and July, and temperatures of 24.1 °C in August were extreme. In the 2021 growing season of maize, which lasted 124 days, there was only 194 mm of rain. Irrigation started on 13 June, significantly earlier than in the previous two years. The amount of water added by irrigation was 240 mm, 210 mm, and 210 mm on the $I_1$, $I_2$, and $I_3$ variants, respectively (Table 6, Figure 5). In the period of May–August, there was 175.3 mm of rain, 86.7 mm less than the LTA. The mean air temperature was 21 °C or 0.5 °C higher in comparison with the long-term average. Rainfall of 16.1 mm and 14.0 mm in May and June and a temperature of 25.4 °C in July had an extreme impact on maize production in the region (Figure 2).

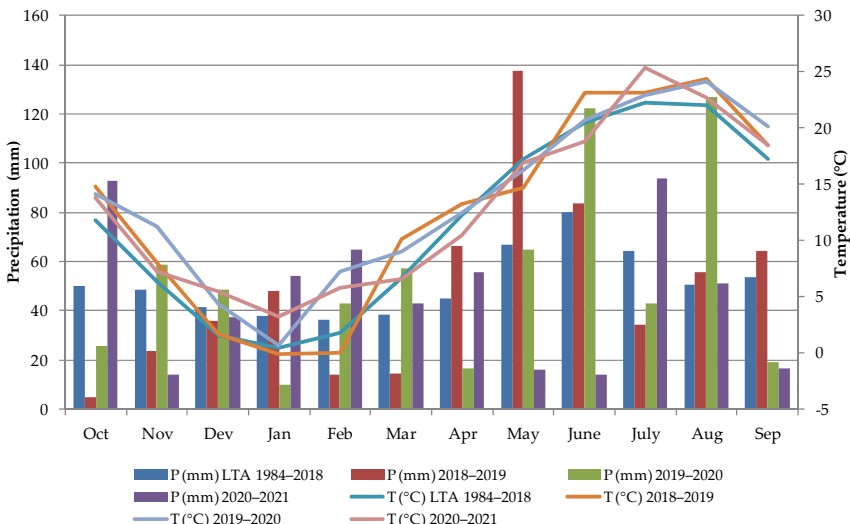

**Figure 2.** Weather data for hydrological years 2018/2019, 2019/2020, and 2020/2021. Bars represent the long-term average (LTA) and the monthly sums of precipitation (P); lines represent long-term (LTA) and monthly average air temperature data (T).

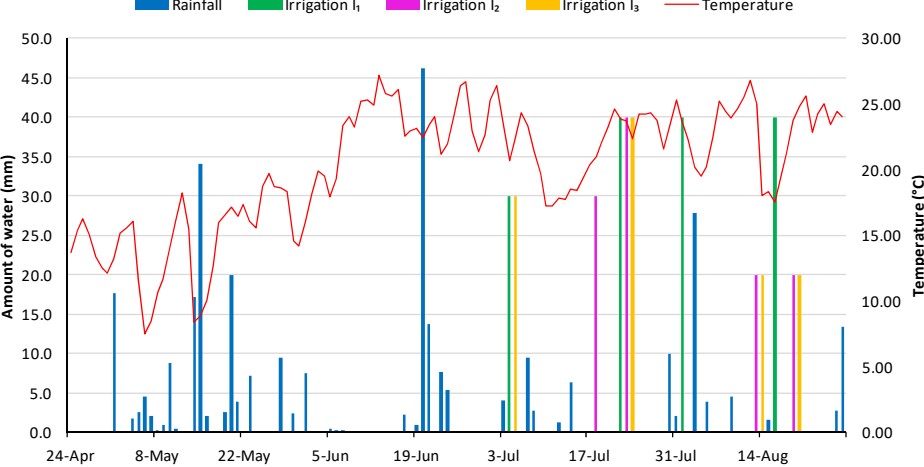

**Figure 3.** Daily weather data (mean air temperature (°C), and rainfall (mm)) and irrigation water applied (reference evapotranspiration ($I_1$), pan evaporation ($I_2$), and local hydrophytothermal indexes ($I_3$)) in 2019.

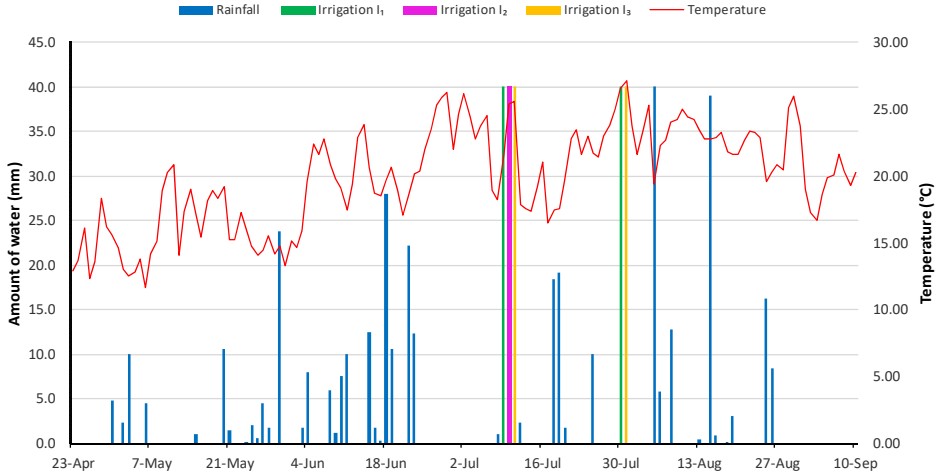

**Figure 4.** Daily weather data (mean air temperature (°C), and rainfall (mm)) and irrigation water applied (reference evapotranspiration ($I_1$), pan evaporation ($I_2$), and local hydrophytothermal indexes ($I_3$)) in 2020.

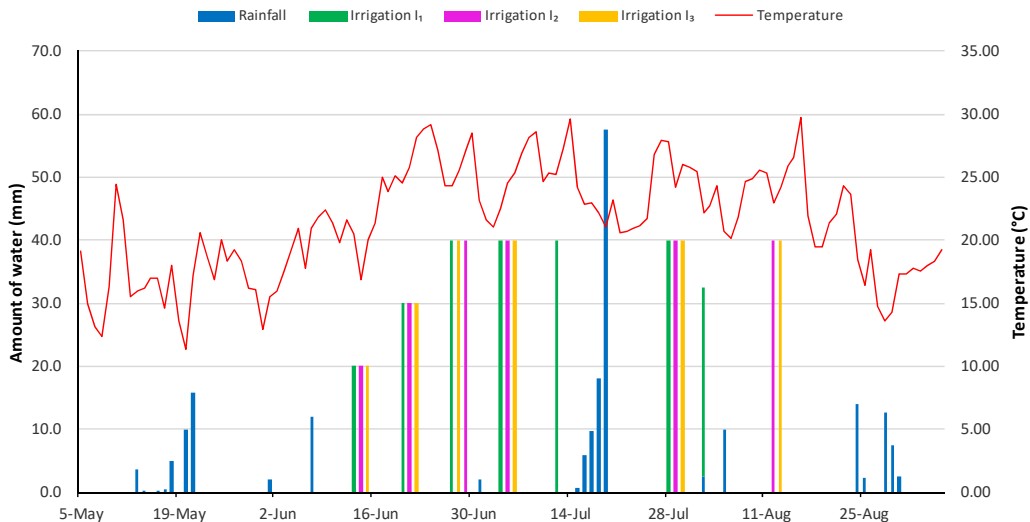

**Figure 5.** Daily weather data (mean air temperature (°C), and rainfall (mm)) and irrigation water applied (reference evapotranspiration ($I_1$), pan evaporation ($I_2$), and local hydrophytothermal indexes ($I_3$)) in 2021.

*3.2. Effect of Irrigation on Yield and Selected Yield Components of Maize*

In all three years, irrigated treatments ($I_1$, $I_2$, and $I_3$) had statistically higher yields and values of selected yield components than the rainfed, nonirrigated variant ($I_o$), except for the weight of 1000 grains in 2020 (Table 3).

3.2.1. Effect of Different ET-Based Irrigation Scheduling on Yield and Selected Yield Components of Maize

Concerning the yield of maize and yield components, $I_1$ is the most acceptable, followed by $I_2$, and finally, $I_3$ (Table 3).

3.2.2. Effect of Surface and Shallow Subsurface Drip Irrigation on Yield and Selected Yield Components of Maize

Statistically significant differences in maize yield between $S_1$ and $S_2$ treatments were not found in the study period. In 2020, the values of ear weight, number of grains per ear, and weight of grains per ear had statistically higher values on the $S_1$ compared to the $S_2$ variant (Table 3).

*3.3. Crop Water Use on Maize Evapotranspiration, Seasonal and Daily Values*

In 2019, the seasonal evapotranspiration values of maize in irrigation conditions ($ET_m$) were 461 mm ($I_1$), 411 mm ($I_2$), 447 mm ($I_3$), and 323–334 mm for the nonirrigated ($I_o$) and control variant ($ET_a$) (Table 4). The highest water used on $ET_m$ was recorded from silking to physiological maturity (VS-R6), which amounted to 197–214 mm or 44.1–47.9% of the total water used during the growing season in the irrigated treatments, but 110–144 mm or 34.1–44.4% from 7–8 pairs of leaves to silking (V7-8-VS) on the rainfed variant. The average seasonal daily evapotranspiration ($ET_d$) varied from 3.6 to 3.7 mm, but the highest average value of daily evapotranspiration ($ET_d$) from 4.5–4.9 mm was detected from VS-R6 (Table 4). A maximum $ET_d$ of 7.5 mm, 6.0 mm, and 5.6 mm was detected on 1 July, 22 July, and 12 August for $I_1$, $I_2$, and $I_3$ variants, respectively (Figure 6).

**Table 3.** Effects of surface and shallow subsurface drip irrigation and different ET-based irrigation scheduling on yield and selected yield components of maize.

| | 2019 | | | | | 2020 | | | | | 2021 | | | | |
|---|---|---|---|---|---|---|---|---|---|---|---|---|---|---|---|
| | $I_1$ | $I_2$ | $I_3$ | $I_0$ | Mean | $I_1$ | $I_2$ | $I_3$ | $I_0$ | Mean | $I_1$ | $I_2$ | $I_3$ | $I_0$ | Mean |
| **Yield (t ha$^{-1}$)** | | | | | $I_1$–$I_3$ | | | | | $I_1$–$I_3$ | | | | | $I_1$–$I_3$ |
| $S_1$ | 10.89 [a] | 8.38 [c] | 8.75 [c] | 7.28 [d] | 9.34 [A] | 12.89 [a] | 11.11 [bc] | 12.20 [ab] | 10.54 [c] | 12.07 [A] | 9.40 [b] | 9.57 [b] | 8.98 [b] | 5.98 [c] | 9.32 [B] |
| $S_2$ | 10.57 [a] | 9.48 [b] | 8.56 [c] | 7.28 [d] | 9.54 [A] | 11.05 [bc] | 11.13 [bc] | 11.85 [abc] | 10.54 [c] | 11.34 [A] | 10.20 [ab] | 10.90 [a] | 8.75 [b] | 5.98 [c] | 9.95 [A] |
| Mean | 10.73 [A] | 8.93 [B] | 8.65 [B] | 7.28 [C] | | 11.97 [A] | 11.12 [AB] | 12.02 [A] | 10.54 [B] | | 9.80 [A] | 10.24 [A] | 8.86 [C] | 5.98 [D] | |
| LSD$_{(0.05)}$ | ET 0.4 | S 0.2 | ET × S 0.5 | | | ET 1.1 | S 0.8 | ET × S 1.6 | | | ET 0.46 | S 0.52 | ET × S 0.86 | | |
| **Weight of ear (g)** | | | | | | | | | | | | | | | |
| $S_1$ | 331.8 [a] | 284.4 [b] | 280.5 [b] | 237.3 [c] | 298.9 [A] | 336.4 [a] | 335.2 [a] | 340.2 [a] | 294.7 [d] | 337.3 [A] | 299.5 [a] | 314.9 [a] | 312.4 [a] | 234.9 [b] | 290.4 [A] |
| $S_2$ | 305.4 [b] | 282.6 [b] | 283.9 [b] | 237.3 [c] | 290.6 [A] | 317.5 [bc] | 329.0 [ab] | 307.4 [cd] | 294.7 [d] | 318.0 [B] | 298.3 [a] | 302.6 [a] | 294.0 [a] | 234.9 [b] | 282.4 [A] |
| Mean | 318.6 [A] | 283.5 [B] | 282.2 [B] | 2.373 [C] | | 326.9 [A] | 332.1 [A] | 323.8 [A] | 294.7 [B] | | 298.9 [A] | 308.7 [A] | 303.2 [A] | 234.9 [B] | |
| LSD$_{(0.05)}$ | ET 23.3 | S 8.7 | ET × S 26.1 | | | ET 8.8 | S 10.5 | ET × S 17.2 | | | ET 19.6 | S 14.3 | ET × S 27.9 | | |
| **Number of grains per ear** | | | | | | | | | | | | | | | |
| $S_1$ | 580 [a] | 551 [ab] | 584 [a] | 529 [b] | 572 [A] | 815 [a] | 768 [ab] | 773 [ab] | 706 [c] | 785 [A] | 679 [a] | 692 [a] | 673 [a] | 598 [b] | 681 [A] |
| $S_2$ | 558 [ab] | 544 [ab] | 570 [ab] | 529 [b] | 557 [A] | 754 [bc] | 754 [bc] | 734 [bc] | 706 [c] | 747 [B] | 635 [ab] | 693 [a] | 648 [ab] | 598 [b] | 658 [A] |
| Mean | 569 [A] | 548 [AB] | 577 [A] | 529 [B] | | 785 [A] | 761 [AB] | 754 [AB] | 706 [B] | | 657 [A] | 692 [A] | 661 [A] | 598 [B] | |
| LSD$_{(0.05)}$ | ET 40.1 | S 19.4 | ET × S 48.1 | | | ET 55.5 | S 14.9 | ET × S 59.0 | | | ET 44.9 | S 28.3 | ET × S 59.5 | | |
| **Weight of grains per ear (g)** | | | | | | | | | | | | | | | |
| $S_1$ | 277.7 [a] | 236.6 [bc] | 235.1 [bc] | 199.0 [d] | 249.8 [A] | 273.8 [a] | 271.7 [a] | 275.7 [a] | 239.2 [c] | 273.7 [A] | 261.2 [a] | 263.6 [a] | 264.1 [a] | 201.0 [b] | 247.5 [A] |
| $S_2$ | 255.8 [b] | 238.4 [bc] | 233.9 [c] | 199.0 [d] | 242.7 [A] | 257.2 [b] | 267.0 [ab] | 249.1 [bc] | 239.2 [c] | 257.8 [B] | 260.5 [a] | 258.5 [a] | 255.8 [a] | 201.0 [b] | 243.9 [A] |
| Mean | 266.7 [A] | 237.5 [B] | 234.5 [B] | 199.0 [C] | | 265.5 [AB] | 269.3 [A] | 262.4 [B] | 239.2 [C] | | 260.8 [A] | 261.1 [A] | 260.0 [A] | 201.0 [B] | |
| LSD$_{(0.05)}$ | ET 19.3 | S 7.2 | ET × S 21.6 | | | ET 6.8 | S 7.9 | ET × S 13.0 | | | ET 28,5 | S 4.2 | ET × S 29.0 | | |
| **Weight of 1000 grains (g)** | | | | | | | | | | | | | | | |
| $S_1$ | 504.7 [a] | 444.6 [b] | 415.0 [cd] | 388.5 [d] | 438.2 [A] | 350.6 [a] | 364.5 [a] | 371.9 [a] | 350.6 [a] | 362.3 [A] | 440.4 [a] | 429.3 [ab] | 409.7 [ab] | 371.4 [b] | 412.7 [A] |
| $S_2$ | 485.0 [a] | 450.6 [b] | 423.5 [bc] | 388.5 [d] | 436.9 [A] | 358.2 [a] | 364.0 [a] | 360.8 [a] | 350.6 [a] | 361.0 [A] | 401.5 [ab] | 408.7 [ab] | 415.3 [ab] | 371.4 [b] | 399.2 [A] |
| Mean | 494.9 [A] | 447.6 [B] | 419.3 [C] | 388.5 [D] | | 354.4 [A] | 364.2 [A] | 366.4 [A] | 350.6 [A] | | 420.9 [A] | 419.0 [AB] | 412.5 [AB] | 371.4 [B] | |
| LSD$_{(0.05)}$ | ET 18.1 | S 14.8 | ET × S 27.4 | | | ET 20.0 | S 8.9 | ET × S 23.4 | | | ET 49.0 | S 25.1 | ET × S 59.9 | | |

Uppercase letters indicate the statistical difference between treatments (ET and S) and lowercase letters represent the interaction between treatments (ET × S). Values followed by different letters, for the same year, are significantly different, but the same letters indicate the absence of statistical significance at the probability level of 0.05 ($p < 0.05$).

**Table 4.** Water balance of maize in 2019.

| Elements | From Sowing to Emergence (S-E) | | | From Emergence to 7–8 Leaves (E-V7-8) | | | From 7–8 Leaves to Silking (V7-8-VS) | | | From Silking to Physiological Maturity (VS-R6) | | | The Entire Season Total/Average (S-R6) | | |
|---|---|---|---|---|---|---|---|---|---|---|---|---|---|---|---|
| | 24.IV–03.V | | | 04.V–14.VI | | | 15.VI–14.VII | | | 15.VII–27.VIII | | | 24.IV–27.VIII | | |
| | $I_1$ | $I_2$ | $I_3$ | $I_1$ | $I_2$ | $I_3$ | $I_1$ | $I_2$ | $I_3$ | $I_1$ | $I_2$ | $I_3$ | $I_1$ | $I_2$ | $I_3$ |
| $ET_o$ (mm) | 39 | - | - | 159 | - | - | 156 | - | - | 225 | - | - | 579 | - | - |
| $E_o$ (mm) | - | 38 | - | - | 160 | - | - | 151 | - | - | 289 | - | - | 638 | - |
| $ET_m$ (mm) | 13 | 16 | 18 | 90 | 88 | 110 | 144 | 110 | 122 | 214 | 197 | 197 | 461 | 411 | 447 |
| $ET_m$ (%) | 2.8 | 3.9 | 4.0 | 19.5 | 21.4 | 24.6 | 31.2 | 26.8 | 27.3 | 46.5 | 47.9 | 44.1 | 100 | 100 | 100 |
| Duration (days) | | 10 | | | 42 | | | 30 | | | 44 | | | 126 | |
| $ET_d$ (mm) | 1.3 | 1.6 | 1.8 | 2.1 | 2.1 | 2.6 | 4.8 | 3.7 | 4.1 | 4.9 | 4.8 | 4.5 | 3.7 | 3.7 | 3.6 |
| Rainfall (mm) | | 18 | | | 129 | | | 100 | | | 66 | | 313 | 313 | 313 |
| Temperature (°C) | | 14.3 | | | 17.1 | | | 22.4 | | | 22.8 | | 20.1 | 20.1 | 20.1 |
| Δ | −3 | −6 | - | +39 | −15 | +19 | −44 | −10 | −22 | −10 | −44 | −18 | - | - | - |
| r (mm) | 21 | 21 | 21 | 18 | 15 | 21 | 54 | 54 | 40 | 10 | 44 | 18 | - | - | - |
| $ET_a$ (mm) | 13 | 16 | 18 | 90 | 88 | 110 | 144 | 110 | 122 | 76 | 110 | 84 | 323 | 324 | 334 |
| d (mm) | 0 | 0 | 0 | 0 | 0 | 0 | 0 | 0 | 0 | 138 | 87 | 113 | 138 | 87 | 113 |
| s (mm) | 0 | 0 | 0 | 3 | 2 | 0 | 0 | 0 | 0 | 0 | 0 | 0 | 3 | 2 | 0 |
| Irrigation (mm) | - | - | - | - | - | - | 30 | - | 30 | 120 | 110 | 80 | 150 | 110 | 110 |

$ET_o$—reference evapotranspiration (mm), $E_o$—pan evaporation (mm), $ET_m$—evapotranspiration in irrigated treatments (mm), $ET_d$—daily evapotranspiration (mm), Δ—a difference in rainfall and $ET_m$, d—water deficit (mm), s—water sufficient (mm), r—soil water reserve at the beginning of the growing season (mm) of RAW, $ET_a$—actual evapotranspiration, rainfed (mm).

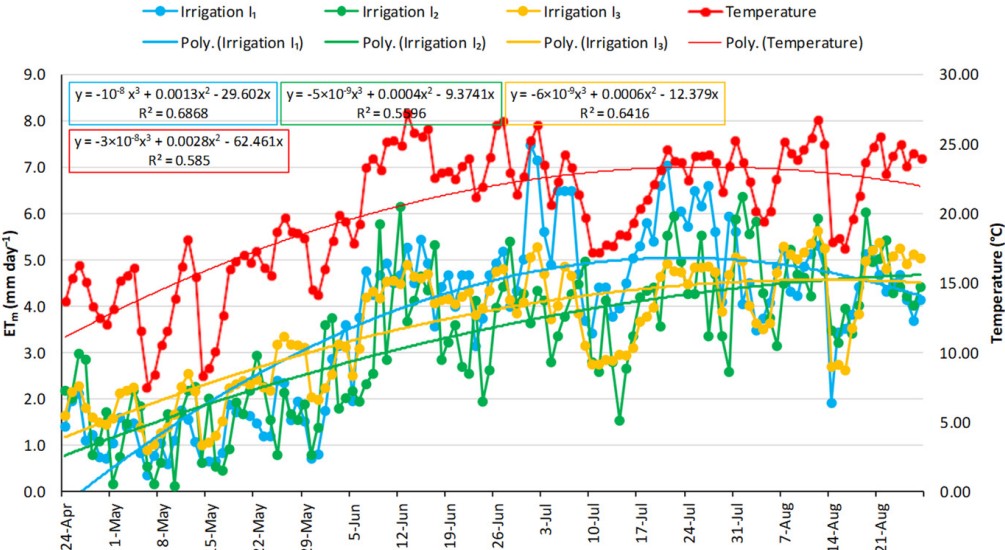

**Figure 6.** Daily maize evapotranspiration in 2019.

In 2020, the seasonal evapotranspiration values of maize in irrigation conditions ($ET_m$) were 478 mm ($I_1$), 430 mm ($I_2$), 481 mm ($I_3$), and 377–397 mm for the nonirrigated ($I_o$), control treatment ($ET_a$) (Table 5). The highest water used on ETm was recorded from VS-R6, which amounted to 205–249 mm or 47.7–52.1% of the total water used during the entire growing season in irrigated treatments and 179 mm or 45.1.5–47.5% on the control treatment. The average season daily evapotranspiration ($ET_d$) varied from 3.1 to 3.4 mm, but the highest average value of daily evapotranspiration ($ET_d$) from 4.4–5.4 mm was detected from V7-8-VS (Table 5). A maximum ETd of 6.9 mm, 6.2 mm, and 5.4 mm was detected on 11 July, 8 June, and 31 July for $I_1$, $I_2$, and $I_3$ treatments, respectively (Figure 7).

**Table 5.** Water balance of maize in 2020.

| Elements | From Sowing to Emergence (S-E) | | | From Emergence to 7–8 Leaves (E-V7-8) | | | From 7–8 Leaves to Silking (V7-8-VS) | | | From Silking to Physiological Maturity (VS-R6) | | | The Entire Season Total/Average (S-R6) | | |
|---|---|---|---|---|---|---|---|---|---|---|---|---|---|---|---|
| | 23.IV–02.V | | | 03.V–25.VI | | | 26.VI–11.VII | | | 12.VII–10.IX | | | 23.IV–10.IX | | |
| | $I_1$ | $I_2$ | $I_3$ | $I_1$ | $I_2$ | $I_3$ | $I_1$ | $I_2$ | $I_3$ | $I_1$ | $I_2$ | $I_3$ | $I_1$ | $I_2$ | $I_3$ |
| $ET_o$ (mm) | 36 | - | - | 222 | - | - | 87 | - | - | 280 | - | - | 625 | - | - |
| $E_o$ (mm) | - | 42 | - | - | 234 | - | - | 100 | - | - | 311 | - | - | 687 | - |
| $ET_m$ (mm) | 13 | 18 | 19 | 130 | 135 | 145 | 86 | 72 | 71 | 249 | 205 | 246 | 478 | 430 | 481 |
| $ET_m$ (%) | 2.7 | 4.2 | 4.0 | 27.2 | 31.4 | 30.1 | 18.0 | 16.7 | 14.8 | 52.1 | 47.7 | 51.1 | 100 | 100 | 100 |
| Duration (days) | | 10 | | | 54 | | | 16 | | | 61 | | | 141 | |
| $ET_d$ (mm) | 1.3 | 1.8 | 1.9 | 2.4 | 2.5 | 2.7 | 5.4 | 4.5 | 4.4 | 4.1 | 3.4 | 4.0 | 3.4 | 3.1 | 3.4 |
| Rainfall (mm) | | 7 | | | 183 | | | 1 | | | 179 | | 370 | 370 | 370 |
| Temperature (°C) | | 14.6 | | | 11.7 | | | 23.7 | | | 21.7 | | 19.9 | 19.9 | 19.9 |
| Δ | −6 | −11 | −12 | +53 | +48 | +38 | −54 | −54 | −53 | - | - | - | - | - | - |
| r (mm) | 27 | 27 | 27 | 21 | 16 | 15 | 54 | 54 | 53 | 0 | 0 | 0 | - | - | - |
| $ET_a$ (mm) | 13 | 18 | 19 | 130 | 135 | 145 | 55 | 55 | 54 | 179 | 179 | 179 | 377 | 387 | 397 |
| d (mm) | 0 | 0 | 0 | 0 | 0 | 0 | 31 | 17 | 17 | 70 | 26 | 67 | 101 | 43 | 84 |
| s (mm) | 0 | 0 | 0 | 20 | 10 | 0 | 0 | 0 | 0 | 0 | 0 | 0 | 20 | 10 | 0 |
| Irrigation (mm) | - | - | - | - | - | - | 40 | 40 | 40 | 40 | - | 40 | 80 | 40 | 80 |

$ET_o$—reference evapotranspiration (mm), $E_o$—pan evaporation (mm), $ET_m$—evapotranspiration in irrigated treatments (mm), $ET_d$—daily evapotranspiration (mm), Δ—a difference in rainfall and $ET_m$, d—water deficit (mm), s—water sufficient (mm), r—soil water reserve at the beginning of the growing season (mm) of RAW, $ET_a$—actual evapotranspiration, rainfed (mm).

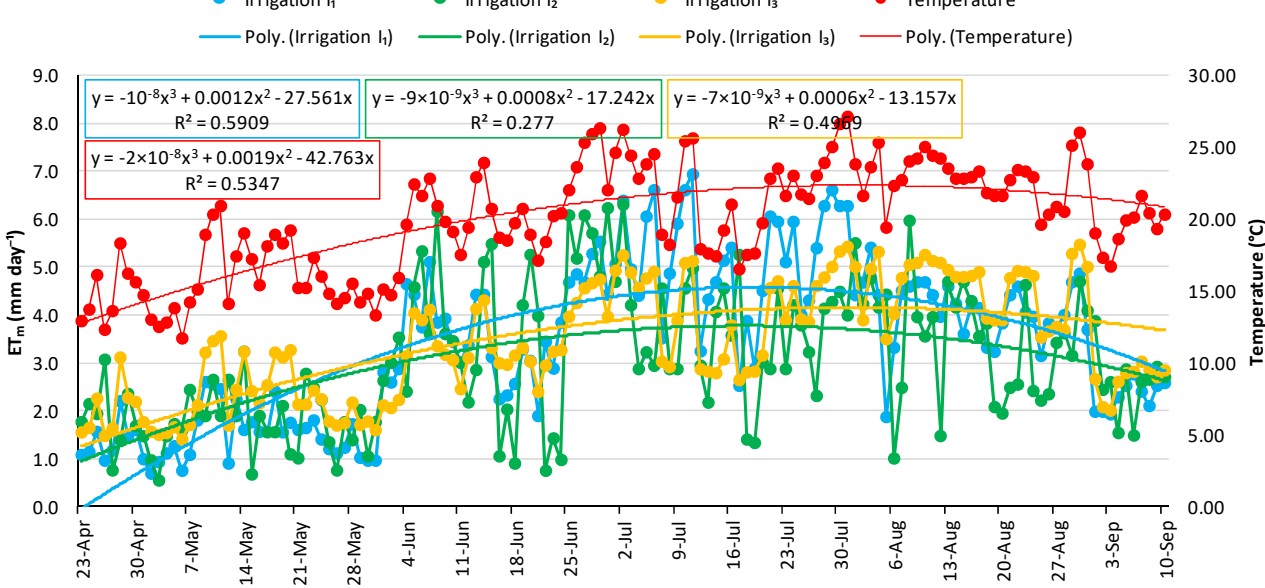

**Figure 7.** Daily maize evapotranspiration in 2020.

In 2021, the seasonal evapotranspiration values of maize in irrigation conditions ($ET_m$) were 514 mm ($I_1$), 473 mm ($I_2$), 471 mm ($I_3$), and 226 mm for the nonirrigated ($I_o$), control treatment ($ET_a$) (Table 6). The highest water used on $ET_m$ was recorded from VS-R6, which amounted to 266–238 mm or 50.5–51.8% of the total water used during the growing season in irrigated treatments and 143 mm or 63.3% on the control treatment, but the average seasonal daily evapotranspiration ($ET_d$) varied from 3.8 to 4.1 mm, but the highest average daily evapotranspiration ($ET_d$) from 4.8–5.8 mm was detected from V7-8-VS (Table 6). A maximum $ET_d$ of 7.9 mm, 7.1 mm, and 6.2 mm was detected on 8 August, 25 June, and 16 August for $I_1$, $I_2$, and $I_3$ treatments, respectively (Figure 8).

**Table 6.** Water balance of maize in 2021.

| Elements | From Sowing to Emergence (S-E) | | | From Emergence to 7–8 Leaves (E-V7-8) | | | From 7–8 Leaves to Silking (V7-8-VS) | | | From Silking to Physiological Maturity (VS-R6) | | | The Entire Season Total/Average (S-R6) | | |
|---|---|---|---|---|---|---|---|---|---|---|---|---|---|---|---|
| | 05.V–11.V | | | 12.V–15.VI | | | 16.VI–10.VII | | | 11.VII–05.IX | | | 05.V–05.IX | | |
| | $I_1$ | $I_2$ | $I_3$ | $I_1$ | $I_2$ | $I_3$ | $I_1$ | $I_2$ | $I_3$ | $I_1$ | $I_2$ | $I_3$ | $I_1$ | $I_2$ | $I_3$ |
| $ET_o$ (mm) | 28 | - | - | 149 | - | - | 155 | - | - | 273 | - | - | 605 | - | - |
| $E_o$ (mm) | - | 32 | - | - | 159 | - | - | 181 | - | - | 352 | - | - | 724 | - |
| $ET_m$ (mm) | 12 | 14 | 18 | 90 | 90 | 96 | 146 | 132 | 119 | 266 | 237 | 238 | 514 | 473 | 471 |
| $ET_m$ (%) | 2.3 | 3.0 | 3.8 | 17.5 | 19.0 | 20.4 | 28.4 | 27.9 | 25.3 | 51.8 | 50.1 | 50.5 | 100 | 100 | 100 |
| Duration (days) | | 7 | | | 35 | | | 25 | | | 57 | | | 124 | |
| $ET_d$ (mm) | 1.7 | 2.0 | 2.6 | 2.6 | 2.6 | 2.7 | 5.8 | 5.3 | 4.8 | 4.7 | 4.2 | 4.2 | 4.1 | 3.8 | 3.8 |
| Rainfall (mm) | | 0 | | | 49 | | | 2 | | | 143 | | 194 | 194 | 194 |
| Temperature (°C) | | 17.4 | | | 17.9 | | | 25.4 | | | 22.3 | | 21.4 | 21.4 | 21.4 |
| Δ | −12 | −14 | −18 | −20 | −18 | −14 | - | - | - | - | - | - | - | - | - |
| r (mm) | 32 | 32 | 32 | 20 | 18 | 14 | 0 | 0 | 0 | 0 | 0 | 0 | - | - | - |
| $ET_a$ (mm) | 12 | 14 | 18 | 69 | 67 | 63 | 2 | 2 | 2 | 143 | 143 | 143 | 226 | 226 | 226 |
| d (mm) | 0 | 0 | 0 | 21 | 23 | 33 | 144 | 130 | 117 | 123 | 94 | 95 | 288 | 247 | 245 |
| s (mm) | 0 | 0 | 0 | 0 | 0 | 0 | 0 | 0 | 0 | 0 | 0 | 0 | 0 | 0 | 0 |
| Irrigation (mm) | - | - | - | 20 | 20 | 20 | 110 | 110 | 110 | 110 | 80 | 80 | 240 | 210 | 210 |

$ET_o$—reference evapotranspiration (mm), $E_o$—pan evaporation (mm), $ET_m$—evapotranspiration in irrigated treatments (mm), $ET_d$—daily evapotranspiration (mm), Δ—a difference in rainfall and $ET_m$, d—water deficit (mm), s—water sufficient (mm), r—soil water reserve at the beginning of the growing season (mm) of RAW, $ET_a$—actual evapotranspiration, rainfed (mm).

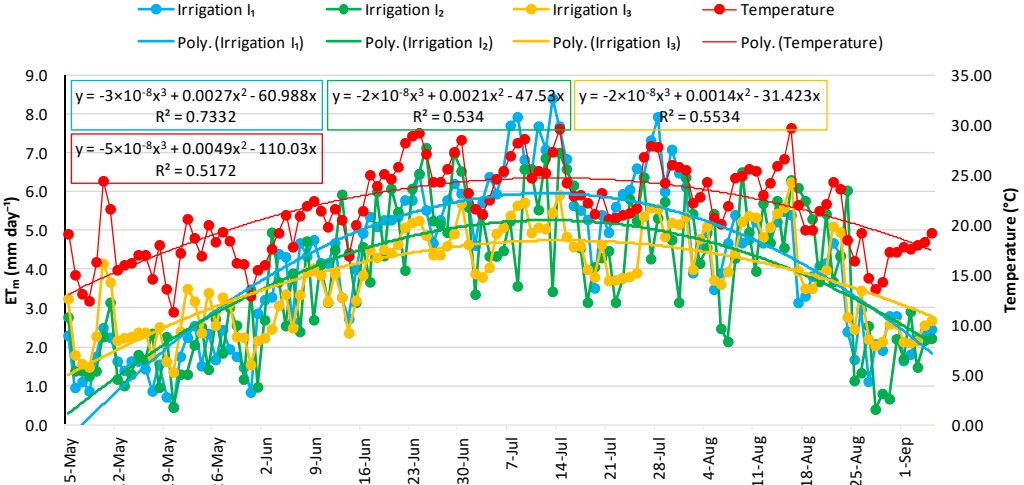

**Figure 8.** Daily maize evapotranspiration in 2021.

### 3.4. Water Use Efficiency and Yield Response Factor of Maize

Statistically significant differences in irrigation and evapotranspiration water use efficiency of maize in the study period were not determined between $S_1$ and $S_2$ or $I_1$, $I_2$, and $I_3$ (Table 7).

**Table 7.** Irrigation and evapotranspiration water use efficiency of maize.

| Year | Type of Drip Irrigation | ET-Based Irrigation Scheduling | IWUE | ETWUE |
|---|---|---|---|---|
| 2019 | $S_1$ | $I_1$ | 2.41 | 2.62 |
| | | $I_2$ | 1.00 | 1.26 |
| | | $I_3$ | 1.34 | 1.30 |
| | $S_2$ | $I_1$ | 2.19 | 2.38 |
| | | $I_2$ | 2.01 | 2.54 |
| | | $I_3$ | 1.17 | 1.14 |

**Table 7.** *Cont.*

| Year | Type of Drip Irrigation | ET-Based Irrigation Scheduling | IWUE | ETWUE |
|---|---|---|---|---|
| 2020 | $S_1$ | $I_1$ | 2.94 | 2.33 |
| | | $I_2$ | 1.42 | 1.32 |
| | | $I_3$ | 2.07 | 1.98 |
| | $S_2$ | $I_1$ | 0.64 | 0.51 |
| | | $I_2$ | 1.48 | 1.38 |
| | | $I_3$ | 1.64 | 1.56 |
| 2021 | $S_1$ | $I_1$ | 1.42 | 1.18 |
| | | $I_2$ | 1.71 | 1.46 |
| | | $I_3$ | 0.71 | 0.61 |
| | $S_2$ | $I_1$ | 1.74 | 1.45 |
| | | $I_2$ | 2.39 | 2.03 |
| | | $I_3$ | 1.58 | 1.36 |
| 2019/2021 | $S_1$ | | 1.67 [a] | 1.56 [a] |
| | $S_2$ | | 1.65 [a] | 1.59 [a] |
| | LSD | | 0.99 | 0.66 |
| | $I_1$ | | 1.89 [a] | 1.74 [a] |
| | $I_2$ | | 1.67 [a] | 1.66 [a] |
| | $I_3$ | | 1.42 [a] | 1.32 [a] |
| | LSD | | 0.76 | 1.08 |

IWUE (Irrigation water use efficiency, kg m$^{-3}$), ETWUE (Evapotranspiration water use efficiency, kg m$^{-3}$), reference evapotranspiration (ET$_o$, $I_1$), pan evaporation (E$_o$, $I_2$), and local hydrophytothermal indexes (l$_c$, $I_3$). The same letters indicate there is no statistically significant difference between treatments within the same column according to the LSD test ($p < 0.05$).

### 3.5. Yield Response Factor

The accuracy of the yield response factor ($K_y$) depends on having a sufficient range and number of values for yield (Y) and evapotranspiration (ET) and assumes that the relationships between Y and ET are linear over this range. In general, relative yield decreased linearly with an increasing relative evapotranspiration deficit. The relationship between maize yield (t ha$^{-1}$) and seasonal crop water use (ET) for the studied period was linear (Figure 9). As an indicator of maize sensitivity to water stress over the three growing seasons, maize $K_y$ was averaged as 0.71 (Figure 10). Statistical differences in $K_y$ values were not found between $S_1$ (0.70) and $S_2$ (0.71), nor between $I_1$ (0.77), $I_2$ (0.72), and $I_3$ (0.63). The weather conditions did not significantly influence the $K_y$ values (Table 8).

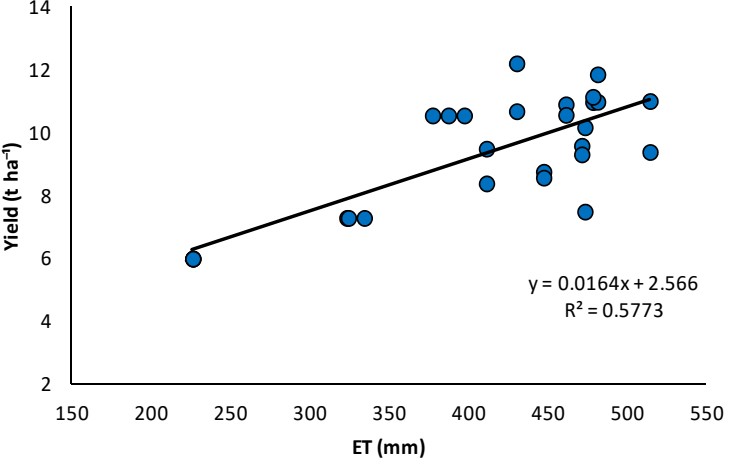

**Figure 9.** Relationship between seasonal evapotranspiration and yield of maize.

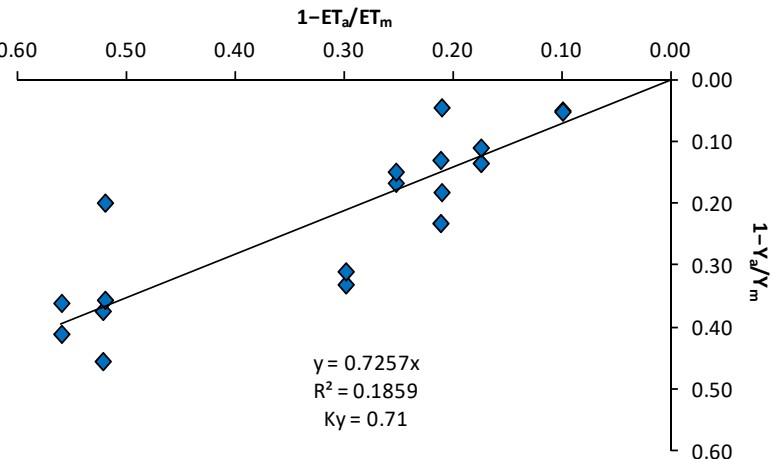

**Figure 10.** Relationship between relative yield decrease ($1 - Y_a/Y_m$) and seasonal relative evapotranspiration deficit ($1 - ET_a/ET_m$) for the three seasons of experimental data combined.

**Table 8.** Yield response factor of maize.

| Year | Type of Irrigation | Irrigation Scheduling | $ET_m$ | $ET_a$ | $1 - ET_a/ET_m$ | $Y_m$ | $Y_a$ | $1 - Y_a/Y_m$ | $K_y$ |
|---|---|---|---|---|---|---|---|---|---|
| 2019 | $S_1$ | $I_1$ | 461 | 323 | 0.299 | 10.894 | | 0.332 | 1.11 |
| | | $I_2$ | 411 | 324 | 0.212 | 8.375 | | 0.131 | 0.62 |
| | | $I_3$ | 447 | 334 | 0.253 | 8.747 | | 0.168 | 0.66 |
| | | $I_0$ | | | | | 7.276 | | |
| | $S_2$ | $I_1$ | 461 | 323 | 0.299 | 10.559 | | 0.311 | 1.04 |
| | | $I_2$ | 411 | 324 | 0.212 | 9.482 | | 0.233 | 1.10 |
| | | $I_3$ | 447 | 334 | 0.253 | 8.558 | | 0.150 | 0.59 |
| | | $I_0$ | | | | | 7.276 | | |
| 2020 | $S_1$ | $I_1$ | 478 | 377 | 0.211 | 12.893 | | 0.183 | 0.87 |
| | | $I_2$ | 430 | 387 | 0.100 | 11.108 | | 0.051 | 0.51 |
| | | $I_3$ | 481 | 397 | 0.175 | 12.199 | | 0.136 | 0.78 |
| | | $I_0$ | | | | | 10.540 | | |
| | $S_2$ | $I_1$ | 478 | 377 | 0.211 | 11.051 | | 0.046 | 0.22 |
| | | $I_2$ | 430 | 387 | 0.100 | 11.133 | | 0.053 | 0.53 |
| | | $I_3$ | 481 | 397 | 0.175 | 11.849 | | 0.111 | 0.63 |
| | | $I_0$ | | | | | 10.540 | | |
| 2021 | $S_1$ | $I_1$ | 514 | 226 | 0.560 | 9.400 | | 0.362 | 0.65 |
| | | $I_2$ | 473 | 226 | 0.522 | 9.574 | | 0.375 | 0.72 |
| | | $I_3$ | 471 | 226 | 0.520 | 7.477 | | 0.200 | 0.39 |
| | | $I_0$ | | | | | 5.980 | | |
| | $S_2$ | $I_1$ | 514 | 226 | 0.560 | 10.200 | | 0.412 | 0.74 |
| | | $I_2$ | 473 | 226 | 0.522 | 10.900 | | 0.456 | 0.84 |
| | | $I_3$ | 471 | 226 | 0.520 | 9.300 | | 0.357 | 0.74 |
| | | $I_0$ | | | | | 5.980 | | |
| | $S_1$ | 0.70 [a] | $I_1$ | | 0.77 [a] | 2019 | | 0.85 [a] | |
| | $S_2$ | 0.71 [a] | $I_2$ | | 0.72 [a] | 2020 | | 0.59 [a] | |
| | LSD | 0.24 | $I_3$ | | 0.63 [a] | 2021 | | 0.68 [a] | |
| | | | LSD | | 0.29 | LSD | | 0.29 | |

$1 - ET_a/ET_m$ is the relative evapotranspiration deficit; $1 - Y_a/Y_m$ is the relative yield decrease; $ET_m$ is the maximum evapotranspiration (mm) corresponding to $Y_m$; $ET_a$ is the actual evapotranspiration (mm) corresponding to $Y_a$; $Y_m$ is the yield under irrigation, (kg ha$^{-1}$); and $Y_a$ is the yield under nonirrigated conditions (kg ha$^{-1}$). The same letters indicate there is no statistically significant difference between treatments within the same column according to the LSD test ($p < 0.05$).

## 4. Discussion

### 4.1. Weather Conditions and Applied Irrigation Water Amount

All three years of the study were different based on the amount and distribution of rainfall. The most favorable for maize production was 2020, with 370 mm of rain in the growing season, followed by 2019, with 319 mm, and the worst was 2021, with only 194 mm. In 2019, irrigation was carried out from 5 July to 16 August; in 2020, irrigation was needed only in July; and in the 2021 growing season, the water deficit appeared at the beginning of June and was present until the third decade of August. Therefore, irrigation in Vojvodina has a supplementary characteristic [5] and is used to supplement precipitation for successful crop production. In the region of supplementary irrigation, unexpected rainfall after irrigation affects the water regime of the soil and the number and schedule of irrigation events on different irrigation variants, which can influence the obtained results of the examined parameters and complicate their interpretation. Zamora-Re et al. [7] also stated that irrigation scheduling is a difficult task for farmers due to the spatial and temporal rainfall variability in Florida. 2020 is the best example to explain the abovementioned statement. The first watering of maize in 2020 was performed on July 10 with 40 mm of water on all irrigation variants. The 18 mm and 19 mm rain that fell on 18 and 19 July, as well as the $ET_d$ calculation, caused the need to water the $I_1$ and $I_3$ variants on 30 and 31 July, and there was no need to water the $I_2$ variant. The favorable distribution and amount of precipitation in 2020 made it possible to achieve high maize yields even in the variant without irrigation (Table 3). Post-watering rainfall of 40 mm and 39 mm on 5 and 15 August, respectively, were sufficient, and irrigation was not required until the end of the maize growing season (Figure 4). All three years were warmer than the LTA; the increase in temperature was 0.9 °C, 0.5 °C, and 0.5 °C for 2019, 2020, and 2021, respectively (Figure 2). In 2019, the warmest year, the vegetation subperiod from silking to physiological maturity lasted only 44 days, compared to the less warm years 2020 and 2021, when this period lasted 61 and 57 days, respectively (Tables 4–6).

### 4.2. Effect of Irrigation on Yield of Maize

In all three years, irrigated treatments ($I_1$, $I_2$, and $I_3$) had statistically higher yields (7.477–12.893 t ha$^{-1}$) and values of selected yield components than the rainfed, nonirrigated treatment ($I_o$) (7.276–10.540 t ha$^{-1}$), except for the weight of 1000 grains in 2020 (Table 3). The results correspond with many studies conducted in different climatic and soil conditions, which confirm that irrigation can positively affect the yield of maize [5,7,38–40]. In Turkey's north-western regions, Çakir [41] highlights a yield of 15 t ha$^{-1}$ under irrigation settings, compared to a yield of 5 t ha$^{-1}$ under rainfed conditions. Significant variations in maize yield under both irrigated and rainfed conditions confirm that the impact of irrigation on an increase in maize yields in the Vojvodina region is influenced by seasonal weather, namely the total amount and distribution of rainfall. The influence of irrigation can be enormous in dry years, whereas it can be moderate or even absent in wet years [5,38].

### 4.3. Effect of Different ET-Based Irrigation Scheduling on Yield of Maize

The highest average seasonal daily evapotranspiration ($ET_d$) in the study period, of 3.7 mm, 3.5 mm, and 3.6 mm, was obtained for the $I_1$, $I_2$, and $I_3$ treatments, respectively. The linear relationship between the yield and ET of maize indicates higher yields with higher values of evapotranspiration (Figure 9). It appears that in the region, priority should be given to reference evapotranspiration ($ET_o$) and crop coefficients ($k_c$) in calculations $ET_d$ for rational maize irrigation in relation to pan evaporation ($E_o$) and local climatic coefficients ($l_c$). The linear relationship between grain yield and ET has also been found by Mengu and Ozgurel [42], Pejić et al. [5], Kusku et al. [43], and Kresović et al. [38]. In contrast, Imrak et al. [20] and Payero et al. [44] reported a nonlinear relationship between yield and ET. Our results agree with the recommendations given by the FAO [9]; the determination of plant water requirements needs to be indirectly calculated through the reference evapotranspiration ($ET_o$). Pejić et al. [32] suggested that both reference evapotranspiration ($ET_o$)

with appropriate crop coefficients ($k_c$) and evaporation from the free water surface ($E_o$) and plant coefficients (k) could be equally used in computing the daily evapotranspiration ($ET_d$) of maize and pepper for irrigation scheduling programs in the climatic conditions of the Vojvodina region. They concluded that priority should be given to $ET_o$ and $k_c$ due to the easy accessibility and reliability of data.

*4.4. Effect of Surface and Shallow Subsurface Drip Irrigation on Yield of Maize*

No statistically significant variations in maize production between the $S_1$ and $S_2$ treatments were observed in the first two years of the study period. In the driest year of 2021, the statistically significant highest yield was determined on $S_2$ treatment (Table 3). This is in agreement with a result of Wu et al. [40] and Wang et al. [45], who did not find statistical differences in the yield of maize between surface and subsurface drip irrigation. However, the obtained results in dry 2021 are consistent with the findings of Sonbol et al. [46] that shallow subsurface drip irrigation is recommended in dry weather to lessen the negative effects of water shortage, boosting significantly good yields. There are very few results in the literature regarding subsurface irrigation with laterals placed shallow in the soil and removed from the field before harvest and used in the following years. To our knowledge, the main advantage of SSDI over SDI is the ability to place the laterals when sowing seeds, allowing plants to emerge evenly and quickly, especially in arid and semi-arid regions. SDI can only be installed after plants have emerged, at a specific stage of plant development, which means that plants must protect the laterals against the influence of the wind [23].

*4.5. Maize Evapotranspiration*

In the study period, the seasonal evapotranspiration of maize in irrigation conditions ($ET_m$) was 461–514 mm ($I_1$), 411–473 mm ($I_2$), 447–481 mm ($I_3$), and 226–397 mm for the nonirrigated control treatment ($ET_a$). Mengu and Ozgurel [42] found maize evapotranspiration of 481.9 mm and 142.1 mm, respectively, with full irrigation and rainfed treatment in the arid climate of western Turkey. Our values correspond with the results of Pejić et al. [4], who reported that the water requirements of maize for the conditions of the Vojvodina region varied from 460 to 530 mm. $ET_m$ of 514 mm ($I_1$) determined in 2021, which was very hot (21.4 °C) and dry (194 mm of rain) in the growing season, is very similar to $ET_m$ of 512 mm detected in 2015 (20.2 °C, 220 mm of rain) [47]. A similar seasonal ET of maize from 470.5 mm to 485.8 mm was determined in the semi-humid climate (476 mm of annual precipitation) of northwestern China with 180 mm of irrigation water applied. ET of 352 mm was recorded at the nonirrigated treatment [45]. ET of maize was determined by Rudnick et al. [39] in the sub-humid/semi-arid climate (469 mm of average growing season precipitation) of Nebraska (midwestern part of the USA), with a nitrogen rate of 196 kg ha$^{-1}$, varying from 472 mm to 550 mm under full irrigated treatment and 410 mm to 485 mm in rainfed settings. Much higher values of maize evapotranspiration, using the local $K_c$ curve, from 634.2 to 697.7 mm for semiarid conditions of northwestern New Mexico, were reported by Djaman et al. [48]. Payero et al. [49] demonstrated that the stated yield versus $ET_c$ relationships for maize are not constant and vary with location, likely because of different rainfall patterns, features of the soil and crop, management strategies, and meteorological conditions from one site to the next. Many authors consider that unpredictable weather conditions in the Vojvodina province, particularly precipitation levels and distribution, create fluctuations in agricultural production [50,51]. The highest $ET_m$ of maize was recorded in the growth period from VS to R6 in the range of 197–214 mm (44.1–47.9%), 205–249 mm (51.1–52.1%), and 237–266 mm (50.1–51.8%) in 2019, 2020, and 2021, respectively. The highest evapotranspiration was recorded from VS to R6 indicating that this is the most sensitive part of the maize growing season regarding water deficits. Wang et al. (2021) reported the same statement; $ET_d$ increased from VE to V12, peaked during VT to R3 stage, and then declined from R3 to R6, indicating VT to R3 as maize's most sensitive period. In rainfed conditions, evapotranspiration of maize ($ET_a$) was recorded in

the growth period from V7-8 to VS in a range of 110–144 mm and 117–144 mm in 2019 and 2021, respectively, and 179 mm in VS to R6 period in 2020. Higher $ET_a$ for rainfed conditions existed during the V7-8 to VS than the VS to R6 period, most likely due to increased leaf senescence late in the growing season due to prolonged water stress [39]. Obtained values are in accordance with the results of Rudnick et al. [39], who recorded very similar results of maize evapotranspiration of 228–253 mm and 233–238 mm, 284–297 mm, and 187–218 mm in vegetative and reproductive periods of the growing season, in irrigated and rainfed conditions, respectively, with a nitrogen rate of 196 kg ha$^{-1}$. In the first study year (2019), the highest average daily evapotranspiration ($ET_d$) of 4.5–4.9 mm was detected in the reproductive period, from VS to R6, but in the second (2020) and the third years (2021), the highest average $ET_d$ from 4.8 to 5.4 mm was recorded in the vegetative period, from V7-8 to VS (Tables 4–6). The average seasonal daily evapotranspiration ($ET_d$) varied from 3.1 to 4.1 mm; however, the highest average daily evapotranspiration ($ET_d$) was detected from 4.5–4.9 mm (VS to R6), 4.4 mm to 5.4 mm (V7-8 to VS), and 4.8–5.8 mm (V7-8 to VS) in 2019, 2020, and 2021, respectively. Djaman et al. [6] reported similar seasonal average $ET_d$ from 3.5 to 3.9 mm under full irrigation treatment in the climate of the Nebraska region in the USA. A maximum maize $ET_d$ of 7.9 mm, 7.1 mm, and 6.2 mm was detected on 8 August (VS-R6), 25 June (V7-8-VS), and 16 August (VS-R6) for $I_1$, $I_2$, and $I_3$ treatments, respectively (Figures 6–8).

*4.6. Water Use Efficiency and Yield Response Factor of Maize*

The coefficients of irrigation (IWUE) and evapotranspiration (ETWUE) water use efficiency are the best tools for evaluating irrigation efficacy. The IWUE gives a more realistic estimate of irrigation performance, whereas the ETWUE establishes whether the growing season is beneficial for plant production. Pejic et al. [47] pointed out that while comparing results, particular care should be taken since WUE's computations may differ [26,30,36,52]. Pejic et al. [53] noted that in climates where irrigation is supplemented, the WUE calculation will be different (the calculation also takes into account the yield and evapotranspiration of the nonirrigated treatment [36]) compared to arid regions where agricultural production cannot be carried out under natural water supply conditions (value calculated as the ratio of yield to the amount of water added from irrigation or the amount of water used in evaporation by the plant [52]). The results obtained under specific soil and climatic conditions can only be compared over the same period because the genetic yield potential was lower in the past; however, cultivation techniques have also been improved [54]. Statistically significant differences in IWUE in the study period were not determined either between $S_1$ (1.67 kg m$^{-3}$) and $S_2$ (1.65 kg m$^{-3}$) nor between $I_1$ (1.89 kg m$^{-3}$), $I_2$ (1.67 kg m$^{-3}$), and $I_3$ (1.42 kg m$^{-3}$) (Table 7). In the same region, for the period of 2000–2007, Pejić et al. [5] determined an average value of IWUE of 1.72 kg m$^{-3}$. The results are in line with Mengu and Ozgurel [42], who reported that maize's IWUE at full irrigation treatment ranged from 1.78 to 2.13 kg m-3 in the arid climate of western Turkey and with the findings of Yazar et al. [55] that maize IWUE was 1.61 kg m$^{-3}$ for the Mediterranean climate of Southern Turkey. Howell [30] claims that the parameter of IWUE typically tends to increase with a drop in irrigation if the water shortage does not occur during a single growth phase of a plant. Generally, IWUE is influenced by crop yield potential, irrigation method, proper irrigation scheduling, and climatic characteristics of the region [43]. Similar to IWUE, statistical differences were not found in ETWUE values between $S_1$ (1.56 kg m$^{-3}$) and $S_2$ (1.59 kg m$^{-3}$) or between $I_1$ (1.74 kg m$^{-3}$), $I_2$ (1.66 kg m$^{-3}$), and $I_3$ (1.32 kg m$^{-3}$) (Table 7). Pejić et al. [5] detected maize ETWUE in the range of 0.67 to 2.34 kg m$^{-3}$ with an average value of 1.5 kg m$^{-3}$ in the climate of the Vojvodina region. These results are similar to the findings of Kusku et al. [43], who reported maize ETWUE from 1.6 to 1.72 kg m$^{-3}$ in a sub-humid climate of Turkey. Wang et al. [31] pointed out that crop yield depends on the rate of water use and that all factors that increase yield and decrease water used for ET positively affect the ETWUE. ET is affected by many factors, such as weather parameters, crop characteristics, irrigation scheduling, and field

management [56]. A good linear relationship between the relative evapotranspiration deficit and relative yield decrease was observed by combining data over the three years (Figure 10). The slope of the line in Figure 10 represents that $K_y$ is 0.71, indicating that the crop is tolerant to water deficit, and recovers partially from stress, demonstrating less-than-proportional reductions in yield with reduced water use. The $K_y$ value obtained in this study is similar to 0.89 reported by Mengu and Ozgurel [42] in the climate of western Turkey, Irmak et al. [20] who reported a $K_y$ of 0.83 in rain year in the climate of the Nebraska region in the USA and 0.92 found by Greaves and Wang [57] in the tropical region of southern Taiwan. On the other hand, our result was lower than the values of $K_y$ obtained by Kipkorir et al. [58] with 1.21, by Cakir [41] with 1.29, by Dagdelen et al. [59] with 1.04, by Oktem [60] with 1.23, and by Kresović et al. [57] with 1.65.

## 5. Conclusions

Based on the results obtained, it can be concluded that supplementary irrigation in temperate climates significantly increases the yield of maize. In relation to the tested parameters, especially the maize yield, reference evapotranspiration ($ET_o$) and crop coefficients ($k_c$) should be recommended as the most acceptable method for assessing maize evapotranspiration in irrigation scheduling. There is no statistically significant difference between irrigation water use efficiency (IWUE) and evapotranspiration water use efficiency (ETWUE), between surface drip (SDI) and shallow subsurface drip irrigation (SSDI), or between different ET-based irrigation scheduling schemes. Preference should be given to SSDI compared to SDI because the installation of laterals can be performed together with the sowing, which can affect the uniform and timely emergence of maize plants. The yield response factor ($K_y$) of 0.71 indicates that maize could be cultivated without irrigation in Vojvodina's moderate environment; however, high and stable yields can only be obtained under irrigation conditions. These results will lead to more precise planning and effective management of maize irrigation, which will be especially evident in the future due to climate change and the occurrence of extreme temperatures and periods of drought in the area.

**Author Contributions:** Conceptualization, B.P. and G.B.; methodology, B.P.; field, laboratory work and software, D.S., I.B. and B.V.; writing original draft preparation, B.P. and D.S.; writing—review and editing, K.M. and G.B.; supervision, V.S. All authors have read and agreed to the published version of the manuscript.

**Funding:** This research was supported by the Ministry of Science, Technological Development, and Innovation of the Republic of Serbia, within the framework of a contract on the realization and financing of scientific research work (project number: 451-03-47/2023-01/200117).

**Institutional Review Board Statement:** Not applicable.

**Informed Consent Statement:** Not applicable.

**Data Availability Statement:** The data presented in this study are available upon request from the corresponding author.

**Conflicts of Interest:** The authors declare no conflict of interest.

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
