# Peer review of "Effect of Different ET-Based Irrigation Scheduling on Grain Yield and Water Use Efficiency of Drip Irrigated Maize"

_agriculture, doi:10.3390/agriculture13101994_

Round 1
Reviewer 1 Report
The manuscript provides specific knowledge how to optimize irrigation of maize based on its water demand predicted by means of different ET calculation methods and using two types of micro-irrigation systems, even this content is not described well either in the title or in the methodology section. Both must be revised.
The increasing demand for understanding how the meteorological and hydrological factors determine crop yield justifies the actuality of the study (even this is not highlighted in the manuscript clearly).
The manuscript is based on an intensive study with practical justification, even it also contains several obvious statements that could be deleted focussing more on the novelties found by this research.
The article highlights sufficient number of national and international references relevant to the topic.
The study is quite complete, though I feel weakness in the description of the methodology applied. Some technical terms should be determined more clearly, especially the irrigation variants and the title must be modified accordingly.
Among the results, some are too obvious, they would not need such an intensive study to be determined. I suggest to put more emphasis on the specific results.
The tables and figures are self-explanatory and well edited except for Fig. 3-5.
The discussion of the results is well structured and sufficiently detailed (even too wordy) citing relevant references.
The Conclusion part contains statements based on own results. Nevertheless, I suggest to extend it with another sentence referring to the probable effect of climate change that must be taken into consideration when scheduling irrigation of maize.
I inserted more than one hundred sticky notes with my specific comments, questions and corrections in the pdf file of the manuscript.

I suggest to revise some sentences in terms of English (especially order of words within the sentences and use of technical terms). Proofreading by a native English speaker is also recommended.
Reviewer 2 Report
This study analyzes the influencing factors of corn yield, water use efficiency, and yield from three aspects: actual evapotranspiration, surface and shallow subsurface drip pipes. The study is of great significance for improving the precise planning and efficient irrigation of corn in the region. However, I think a revision should be made before it can be accepted. Below are details comments for the manuscript:
1. In the introduction part, I suggest shortening the first paragraph.
2. Line 411, Suggest starting from 150 in the horizontal axis of Figure 9, and suggest starting from 2 for the vertical axis of Figure 9.
I think minor editing of English language required
Reviewer 3 Report
Maize, as an important food crop in Serbia, is of great significance for its food production. This study was conducted in the specific area of Vojvodina, where crop production requires irrigation to supplement the water required for crops. This study mainly studies the response of maize yield and water use characteristics to actual evapotranspiration through two variables: surface drip irrigation and shallow drip irrigation. The article design is reasonable, the data analysis is real and effective, the literature citation is standardized, and the chart expression is accurate. The conclusions obtained from the study have certain practical significance for maize production in the Vojvodina region of Serbia. Suggest publishing the article after minor revisions, as there are several issues with the article.
1. The article title has too many words. Although the title fully summarizes the research content of this study, I think it has a bit too many words. I suggest refining it.
2. This study was only conducted in Vojvodina, a specific area where maize can still rely on natural precipitation to supplement the water needed for growth and development without irrigation. In the future, it is recommended to conduct research in other major maize producing areas or under controlled conditions in Serbia to further verify the accuracy of the research results.
3. The English writing proficiency is still good, but further polishing and modification are still needed.
4. The first paragraph of the Introduction is recommended to be condensed appropriately, and some common-sense sentences do not need to be overly described.
5. The second paragraph of the Introduction is suggested to be merged with the first paragraph. The purpose of writing these two paragraphs is to introduce the research background and necessity, and the importance of drip irrigation is introduced here. I think it can be combined with the previous paragraph to explain.
6. Lines 188-189, which growth period does topdressing refer to for corn?
7. Lines 250, supplement information on the manufacturer and country of production of the instrument.
The English writing proficiency is still good, but further polishing and modification are still needed.
Round 2
Reviewer 1 Report
The manuscript was significantly improved taking all my recommendations into consideration.
